# Decoding the Atmosphere: Optimising Probabilistic Forecasts with Information Gain

**John R. Lawson** [1,*], **Corey K. Potvin** [2,3] **and Kenric Nelson** [4]

1    Bingham Research Center, Utah State University, Vernal, UT 84078, USA
2    NOAA/OAR/National Severe Storms Laboratory, Norman, OK 73072, USA; corey.potvin@noaa.gov
3    School of Meteorology, University of Oklahoma, Norman, OK 73072, USA
4    Photrek, Watertown, MA 02472, USA; kenric.nelson@photrek.io
*    Correspondence: john@jrl.ac

**Abstract:** Probabilistic prediction models exist to reduce surprise about future events. This paper explores the evaluation of such forecasts when the event of interest is rare. We review how the family of Brier-type scores may be ill-suited to evaluate predictions of rare events, and we offer an alternative to information-theoretical scores such as Ignorance. The reduction in surprise provided by a set of forecasts is represented as information gain, a frequent loss function in machine learning training, meaning the reduction in ignorance over a baseline having received a new forecast. We evaluate predictions of a synthetic dataset of rare events and demonstrate the differences in interpretation of the same datasets depending on whether the Brier or Ignorance score is used. While the two types of scores are broadly similar, there are substantial differences in interpretation at extreme probabilities. Information gain is measured in units of *bits*, an irreducible unit of information, that allows forecasts of different variables to be comparatively evaluated fairly. Further insight from information-based scores is gained via a similar reliability–discrimination decomposition as found in Brier-type scores. We conclude by crystallising multiple concepts to better equip forecast-system developers and decision-makers with tools to navigate complex trade-offs and uncertainties that characterise meteorological forecasting. To this end, we also provide computer code to reproduce data and figures herein.

**Keywords:** verification; risk; rare events; information theory; predictability



## 1. Introduction

Rare events can cause extreme harm to human activity. In reducing uncertainty before a high-impact event to mitigate potential damage to life and property, it is important to optimise probabilistic predictions with the right tools to inform forecast-system development. Herein, we argue that a common evaluation metric, the Brier Score (BS) [1], may not be mathematically appropriate for evaluating a dataset containing rare events. The need for information-theoretical metrics as the alternative has been addressed in multiple papers [2–8]. However, the use of logarithmic-based scores such as Ignorance [2] (IGN) is still less common than the BS in meteorology. Below, we review studies that critique and develop alternative scoring rules, and therefrom promote a framework of evaluation techniques and visualisations that use the concept of *information* in a manner familiar to meteorologists. We aim to improve the accessibility of the information-theory framework to academics, forecasters, and evaluators when evaluating and optimising probabilistic predictions.

Probabilistic forecasting systems include traditional numerical weather-prediction (NWP) models, machine learning and artificial intelligence, or less complex statistical models reviewed in [9]. No matter the variable or event, a prediction reduces to evaluating a forecast probability with a posterior probability of occurrence: whether an event was observed or not, which is not always certain. This brings up the spectre of observational uncertainty, often ignored in verification studies by assuming observations are infallible.

In this paper, we promote the idea of measuring information that has been gained from a baseline [10], such as a probability gained from a previous model run, or a climatological forecast. This ignores the problem of observational error altogether. There are ways to include observational error explicitly in some scores [11,12], but the characteristics of observational error are not always known. The error inherent in both the chaotic flow and sensor-derived observations is not considered when evaluating the performance of one model over another for the same events: this lets the evaluator skirt the issue of observational uncertainty more gracefully. Ultimately, the scoring rule we seek is used to measure the degree of surprise. For example, a human forecaster updates their internal confidence according to the receipt of new model guidance; this surprise manifests as a shift in probability distribution before and after the forecast is in hand [13]. This is a similar concept to the use of win probabilities used in sports broadcasting [14] and how large swings may indicate a headline event (and not necessarily a poor forecasting model).

We first cover the background of Brier-family and Information-based scores in Section 2 and their suitability for evaluating rare events in Section 3. The concepts of reliability and discrimination are discussed in Section 4. We discuss the behaviour of these scores by evaluating a synthetic dataset as a case study in Section 5, including visualisations and interpretations accessible to meteorologists familiar with scoring rules like the BS. We synthesise the results and conclude in Section 6. For readability, we moved longer stretches of procedural mathematics to Appendixes A–D. All figures and code used to produce this manuscript are included as python code in Data Availability Statement.

## 2. Background: Brier Score and Information

Readers may find a wider summary of verification metrics in texts such as [13,15–18]. Evaluation of probabilistic forecasts in meteorology is commonly performed with scoring rules that punish the "distance" between a forecast and observation probability ($f$ and $o$, respectively). For the BS, the distance is measured with the mean-square error (MSE):

$$\text{BS} = (f - o)^2 \tag{1}$$

which is typically given as a mean-average over a time series of forecasts $f \in [0,1]$ [1]; observational error is not considered, i.e., $o \in \{0,1\}$. The BS evaluates probabilistic forecasts with the use of mean-square-error (MSE) between forecast distributions and whether the event occurred or not: unity and zero, respectively. This makes the score dimensionless and symmetric: the mean-squared difference is not sensitive to the order of $f$ and $o$. It forms the basis of a commonly used method to verify spatial forecasts, the Fractions Skill Score [19] and its probabilistic equivalents [20,21]; these scores inherit some characteristics of BS:

- The BS gives a disproportionate weight to common events.
- The units of the BS are dimensionless, rendering interpretation straightforward but making the comparison between predictions of different variables difficult and subjective.
- The application of MSE to measure the distance between two pdfs is constrained by the range of probabilities $[0,1]$, imposing a geometric behaviour on the BS function different from the typical interpretation of MSE as between unbounded scalars.
- Overconfident forecasts of rare events that do not occur are punished less severely than with a logarithmic score [4,10].

In contrast, in fields such as machine learning, scoring rules based on information theory are more commonly used [22–25], though not universally [26]. In the information-theory framework, we measure the amount of uncertainty about an event yet to occur, termed *self-entropy*:

$$H_S(p) = -\log_2 p \tag{2}$$

where $p$ is the event probability. This is closely related to scores such as IGN [2], the divergence score [27], logarithmic score [28], and cross-entropy [11,13]. This self-entropy is also known as self-information, uncertainty, surprise, ignorance, choice, amongst other terms [13,29]. Hence, we have improvements measured by information gain [10] (IG) as difference in surprise or

ignorance. We seek to maximise information (i.e., a large value of IG) transferred to the end user from a correct forecast, analogous to the reduction in BS for our new model.

## 3. Evaluating Forecast of Rare Events

In a system with binary outcomes, there are two extremes of high uncertainty [29,30], alternatively conceptualised as low predictability [31,32]. One statistical regime of high uncertainty is found when predicting an event that occurs half the time, such as heads in a coin flip. Average uncertainty is at its maximum: the observer is fully naïve before the coin is flipped and should assume a prior probability of 50%. Another is when an event is very rare (<1%): the closer a forecast probability is to zero, the exponentially more surprising (cf. Equation (2)) to the user when the event occurs. The first regime represents the cusp between two states (basins of attraction) that exist when predicting exceedance of the 50th (median) percentile; the second represents difficulty detecting a perceived "noise in the signal" (i.e., outliers in an ensemble), such as predicting exceedance of a high (99th) percentile.

When optimising probabilistic forecasting systems, small gains can be practically valuable in either high-surprise regime discussed above. There exist diminishing returns perfecting such systems near Lorenzian limits of predictability [33] and computational power for both traditional Eulerian and AI-driven weather prediction. We propose optimising a predictive system with a scoring rule that is more mathematically sound for measuring informative guidance, and hence more likely to yield a useful forecast probability than the use of BS. First, we show there is only one score that meets our desired criteria.

### 3.1. Desired Scoring Rule

What properties do we seek in a mathematical evaluation of our prediction system [4,16,18,34]? Let us divide the problems of choosing a scoring rule:

1. What do we wish to quantify regarding the forecast's benefit?
2. To convert a complex set of forecast and observed values into an insightful score, we must shed information (in both the colloquial sense and that of information theory) to reduce the uncertainty distributions to a single or small set of values that convey characteristics of the forecasts. How should we perform this reduction while preserving as much salient information as possible?

Benedetti asserts that probability, in general, represents the degree of belief an event will occur [4]. However, upon meteorological application, if probabilities were assigned as the belief of the forecaster, then one score does not fit all due to the human sub-optimal assessment of confidence that varies from one forecaster to the next [35]. Instead, the standard meteorological concept of probabilistic forecasts is that the probabilities reflect the likelihood of the event (a proxy of confidence): given the conditions, we expect that a, say, $p = 10\%$ risk of an event occurring should verify with the same frequencies (i.e., 1 in 10 forecasts) within a given temporal and spatial range [36]. These probabilities are often created with an ensemble (Monte Carlo) set of numerical integrations that differ slightly in initial conditions and model configuration; the output should be calibrated to optimise evaluation statistics [37] and account for the undersampling of potential future events [38–40]. In this paradigm, a perfect probabilistic forecast should match the notional true distribution of uncertainty, although this is unobtainable in practice—especially considering inextricable uncertainty in the forecast of a complex, chaotic system [41]. Herein, we set problem (1) as assessing the quality of issued forecasts under extreme uncertainty to highlight a divergence between two related and common scoring rules; thus, we address (2) by examining how these two scores behave in this regime.

It is well accepted that appropriate scoring rules require the property of *propriety* [42–44]; a proper score punishes forecasts that hedge, i.e., forecasts that do not match the true uncertainty distribution in order to minimise a perceived alternative error metric (e.g., MSE). Hedging does not leverage the true belief or knowledge of the issuing party and encourages overly safe or biased forecasts. We also prefer *additivity*; in short, all samples are equal in weight in the dataset. Additivity allows us to apply our own importance to granules of information

gain, each from a sample or event. It also lets us evaluate across sample catalogs and variables with a like-for-like comparison. The third property is less frequently discussed [4,45]: we seek a score that considers a probabilistic forecast a function solely of its probability bin and no others: so-called *locality*. This property is required to avoid the score's influence by probabilities assigned to other outcomes, i.e., a reward for being close. The Brier score (and related multi-category scores) is *nonlocal*: the same sequence of forecast–observation pairs may yield different scores due to forecasts assigned to non-observed events. If we accept this as undesirable, then the only score that fulfils these three requirements is generalised [4] as

$$S(x) = a \log_2 p + c \tag{3}$$

where the base of the logarithm fixes the units, $p$ is the evaluated probability, and $a, c$ represent arbitrary constants. Hence, we find the family of information-theoretical scores to be the only sort that satisfies all three criteria [34,44]; compare Equations (2) and (3). Here, we choose logarithms of base-2 to employ the *bit*: Shannon's intrinsic unit of predictability. The uncertainty of heads before one flips a fair coin is one bit; this is resolved once the coin's face is revealed. An observer's guess beforehand is either correct or incorrect (flipped). This single bit represents a choice between two equal outcomes. It is context-free: it has no further inherent meaning than one gram or metre as quantised blocks of measurement.

*3.2. Measuring Skill with Information Gain*

Our score should measure the reduction in uncertainty (in information-theoretical terms, *entropy*) by issuing a new forecast. Equation (2) represents surprise: one flip of a fair coin ($p = 0.5$) yields 1 bit of uncertainty that is resolved upon receiving the result of heads or tails. There is a high average surprise when the base rate is close to 50%; when a rare event occurs with little warning, there is a low average surprise over the time series but a high surprise per event. This trade-off is common for maximising communication efficiency as in the Shannon–Hartley theory [46–48]. We aim to measure forecast skill in terms of how much surprise was reduced after gaining new information through a forecast about a future event. We define information gain as the baseline self-entropy $H_s(b)$ subtracted by the forecast self-entropy $H_s(f)$. Through logarithmic identities and Equation (2),

$$IG = H_S(b) - H_S(f) \tag{4}$$

$$IG = \log_2 \left( \frac{-\log_2 b}{-\log_2 f} \right) = \log_2 \frac{b}{f} \tag{5}$$

From the self-entropy $H_S$ defined in Equation (2), we consider the weighted combination of all outcomes in the set of possible events $\mathcal{X}$ to derive the Kullback–Leibler Divergence $D_{\text{KL}}(P \parallel Q)$ between a baseline distribution $P$ to the forecast distribution $Q$:

$$IG = H_S(b) - H_S(f) \tag{6}$$

$$= \sum_{x \in \mathcal{X}} P(x) \log_2 P(x) - \sum_{x \in \mathcal{X}} P(x) \log_2 Q(x) \tag{7}$$

$$= \sum_{x \in \mathcal{X}} P(x) \log_2 \left( \frac{P(x)}{Q(x)} \right) \tag{8}$$

$$\equiv D_{\text{KL}}(P \parallel Q) \tag{9}$$

This measures a sort of distance between a priori and posterior pdfs; it is also known as *relative entropy* [2,5]. We use the notation above that matches the information theory convention. For further insight into the connection with information gain, we can rewrite $D_{\text{KL}}$ as:

$$D_{\text{KL}}(P \parallel Q) \equiv \mathbb{E}_{x \sim P} \left[ \log_2 \left( \frac{P(x)}{Q(x)} \right) \right] \tag{10}$$

Here, $\mathbb{E}_{x \sim P}$ denotes the *expectation* of the log-ratio term (cf. Equation (4)) over the events represented by $x$. This expectation can be interpreted as the mean score, weighted by the frequency of event $x$ in the "truth" time series of $P$. We then define ignorance to refer to the scoring rule that employs $D_{\text{KL}}$ [2] to measure uncertainty about $o$ remaining after receiving $f$:

$$\text{IGN} = D_{\text{KL}}(o \parallel f) \tag{11}$$

We return and reformulate information gain, using $f_1$ as the old model's baseline forecast and $f_2$ as the new model's forecast (cf. $b$ and $f$ in Equation (5)):

$$\text{IG} = D_{\text{KL}}(o \parallel f_1) - D_{\text{KL}}(o \parallel f_2) \tag{12}$$

$$= \sum_{x \in \mathcal{X}} o(x) \log_2\left(\frac{o(x)}{f_1(x)}\right) - \sum_{x \in \mathcal{X}} o(x) \log_2\left(\frac{o(x)}{f_2(x)}\right) \tag{13}$$

Again, $o \in \{0, 1\}$ represents observational truth. Above, we subtract one score for the new model's forecast $f_2$, represented as $D_{\text{KL}}(o, f_2)$, from that of another model, $D_{\text{KL}}(o, f_1)$, and thence estimate information gained (or lost) by receipt of $f_2$. If we are setting a baseline (i.e., there is no $f_2$), we solely take $D_{\text{KL}}(o \parallel f_1)$ to be the uncertainty that needs to be removed to obtain a perfect forecast (neglecting observation uncertainty). The baseline herein is set as the observation base-rate frequency $\bar{o}$, but could be persistence, another function of the observed base-rate (i.e., entropy or uncertainty of climatology), etc.

### 3.3. Implications of Using Brier Score to Evaluate Predictions of Rare Events

Benedetti states that BS is proportional to a second-order approximation of its information-theoretical brethren through Taylor expansions [4]. A longer discussion is found therein, but for accessibility here, we lay out the full intermediate steps absent from Benedetti's published derivation [4] in Appendix A. Use of the BS, while making the calculation computationally less demanding by using MSE instead of logarithms, neglects higher-order moments beyond the mean and variance. The fourth moment (kurtosis) serves as a gauge of the distribution's "tailedness", where a fatter tail indicates a higher likelihood of extreme outcomes than a distribution with a smaller kurtosis (such as a Gaussian curve with a mean of zero and standard deviation of unity). This underscores a potential limitation of relying solely on BS for assessment of extreme events or outlier predictions. Employing IG and IGN, despite the logarithm's computational overhead, retains nuances in the higher moments about the forecast distribution's shape and tail behaviour.

The mathematical differences between BS and IGN motivate two tasks: (1) evaluate a synthetic time series to better understand the differences in behaviour between the scores, and (2) determine which family of scores may be better suited to our application in evaluating and optimising forecasts of rare events.

## 4. Decomposition into Reliability and Discrimination

We can reveal more aspects of model evaluation by decomposing the information gained into three components. There are commonalities of terminology between sub-disciplines, and in Table 1, we compare synonyms of key terms used herein.

**Table 1.** Synonyms between concepts in meteorology and information theory.

| Term | Synonyms |
|---|---|
| Ignorance | Entropy, Uncertainty, Surprise |
| Discrimination | Resolution, Sharpness, Goodness |
| Reliability | Calibration, Spread, Fit, Confidence |

Surprise comes in three forms within information-based scores. We can form IGN from the components of uncertainty UNC, reliability REL, and discrimination DSC:

$$\text{IGN} = \text{UNC} - \text{DSC} + \text{REL} \tag{14}$$

Full mathematical derivations can be found in [5] and Appendix C; full Python code is also available in Data Availability Statement. Here, we briefly discuss the interpretation: Equation (14) states that ignorance measures how uncertainty is reduced by the discrimination skill of the model, balanced by the surprise an end user experiences having used the probabilistic forecast.

The first component, uncertainty (UNC), is information entropy inherent in the flow; this component cancels when two forecast systems are evaluated over the same event catalogue, and the remaining uncertainty is differenced. Discrimination (DSC) is also known as resolution in meteorology, a term avoided here due to potential confusion with grid-spacing distance in NWP models. This component measures inherent goodness—the ability to categorise a future event (such as occurring or not). Finally, reliability (REL) is a calibration. For a perfectly calibrated system, an event of interest would be observed 1-in-10 times that a forecast of 10% is issued. This is familiar to meteorologists as an insufficient or excessive spread of members in an NWP ensemble. One cannot evaluate a probabilistic forecast's discrimination and reliability with a single data point, as the scalars cannot be decomposed. Further, a small dataset has an insufficient amount of samples to capture the nature of rare events: a problem for IGN and BS alike; a sufficiently long time series is required to generate meaningful components.

Uncertainty is inextricable; in contrast, REL and DSC components can be optimised, but potentially with one at the expense of the other. This parallels the "Doppler dilemma" [49], where Doppler radars must balance increasing pulse-repetition frequency with a consequent reduction in the maximum unambiguous range. In NWP ensembles, a set of members perturbed from a "best guess" (the control or a deterministic run) may yield fewer skillful ensemble members but a better overall probabilistic prediction. In an underdispersive prediction system (i.e., too confident in a smaller range of future states), events occurring at the tail of the underlying climatological distribution are less likely to be adequately detected. This limits the forecast utility when risk thresholds are critical for decision-making and loss mitigation. However, when confidence is spread quasi-uniformly across a wide range of outcomes, the recipient cannot move far from their prior expectation probability with a wide spread of eventualities. The use of DSC and REL components to evaluate probabilistic-forecast performance is found in [50–52], amongst others.

## 5. Visualising Surprise

To compare the behaviour of information- and Brier-based scores for verification, we use IGN to measure the remaining uncertainty in a forecast, acknowledging this symbology is more familiar to meteorologists. Hence, information gained is the removal of ignorance and having the new forecast over the old one, as in Equation (12), analogous to the reduction in Brier Score (as lower scores of both IGN and BS are better). To compare the two scores, we normalise by a baseline (such as a measure of uncertainty, a function of the base rate). A skill score SS quantifies the relative accuracy of a forecast probability $f$ compared to a baseline $b$:

$$\text{SS} = 1 - \frac{f}{b} \tag{15}$$

A positive SS indicates that the forecast $f$ is more accurate than the baseline $b$, while a negative SS suggests that the forecast model possesses less skill compared to the baseline. Skill scores generated using Equation (15) are labelled the ignorance skill score (IGNSS) and Brier Skill Score (BSS).

### 5.1. Worked Example

We present a case study that leverages the analysis above to create insightful visualisations, reveal forecast qualities that evaluators may already use, and show how the interpretation differs or otherwise. We plot the Brier- and Information-family scores together for all figures for comparison where possible. Thus, we compare the characteristics of both scoring rules to deduce which score may be better suited to our requirements. The

process of generating a time series with rare events and forecasting involves several key steps, including the initialisation of observed occurrences (*o*), introducing auto-correlation, generating forecast probabilities (*f*), and ensuring these probabilities remain within valid bounds. These are discussed in Appendix D; code and figures are available in the Data Availability Statement Python Jupyter notebook.

We generate the old and new models with 10,000 forecast–observation pairs, around 3 times more random error in the old model than the new model, and probability bins $K = \{0.005, 0.01, 0.05, 0.10, 0.20, \ldots 0.80, 0.90, 0.95, 0.99, 0.995\}$. The base rate of occurrence $\bar{o}$ is 0.5%. The distributions of our old and new models are shown in Figure 1. We have deliberately set the old model to be less skillful by introducing more random errors. This is seen in the old model as (1) a larger proportion of near-zero probability forecasts and (2) more forecasts away from the extremes (i.e., less confident in either outcome).

(a)

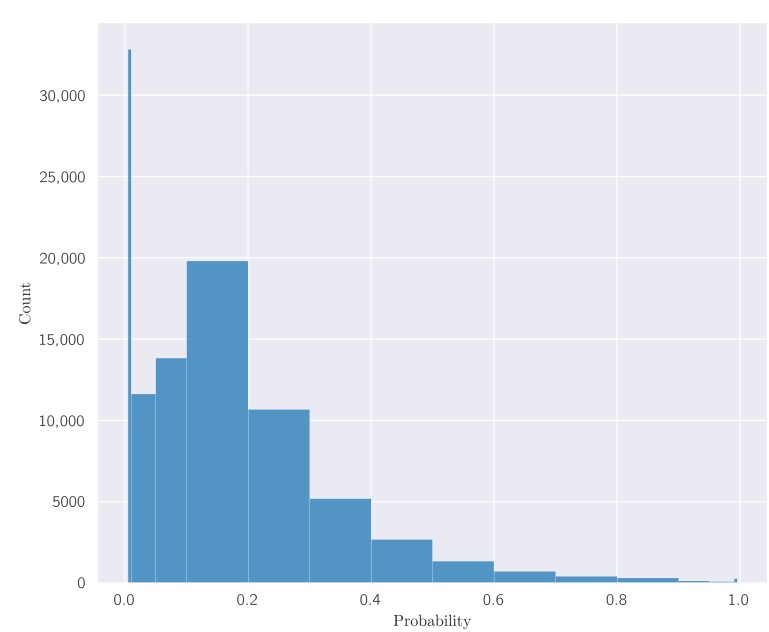

(b)

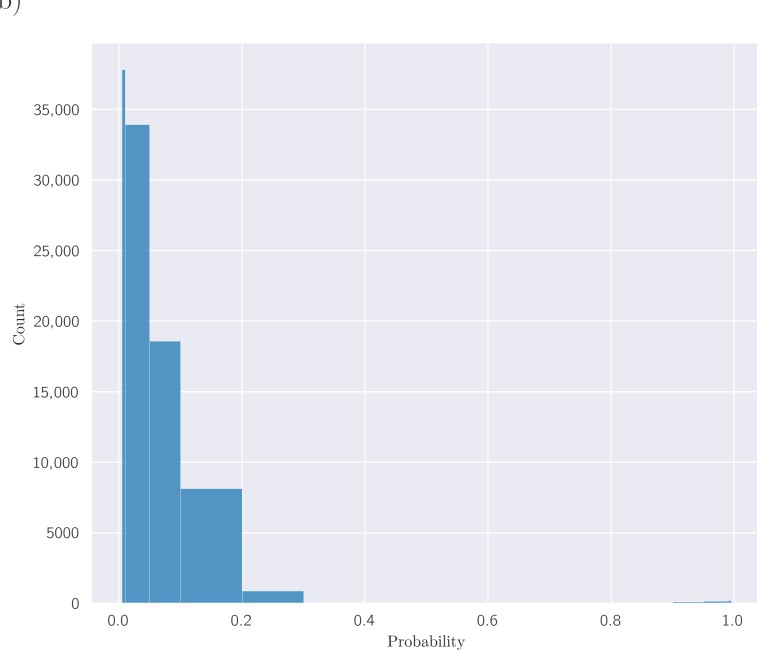

**Figure 1.** Histogram of forecast distribution by probability bin for the old (**a**) and new (**b**) models.

## 5.2. Evaluating Our Test Case

We can take the exponential of negative IGN to yield an average probability issued when an event occurs, which may be a more intuitive way to communicate the forecast performance:

$$P_{\mathrm{avg}} = B^{-\mathrm{IGN}} \tag{16}$$

where $B$ is the IGN logarithm base (here, 2). When appropriate, we show $P_{\mathrm{avg}}$ (as a probability) against IG or IGN (in bits). We find the old and new models left 0.19 and 0.07 bit uncertainty, respectively, yielding 0.12 bit of information gained by the new model. Alternatively, this implies the old model predicted 87.7%, on average, for events that occurred; in contrast, the new model issued 95% for the same events.

We now show a section of a time series of observations and forecasts, taken from the old model at a time when a rare event occurred (Figure 2). During the event occurrence (shaded red in the figure), the old model forecast detects the event's occurrence well (a small difference between the probability and unity), albeit with a small delay, and it also predicts the event lasts too long. Accordingly, the BS value is large in two zones:

1.    The event occurs but was not forecast, as happens very early in the event in our example; and
2.    The event does not occur but was forecast, as happens after the event.

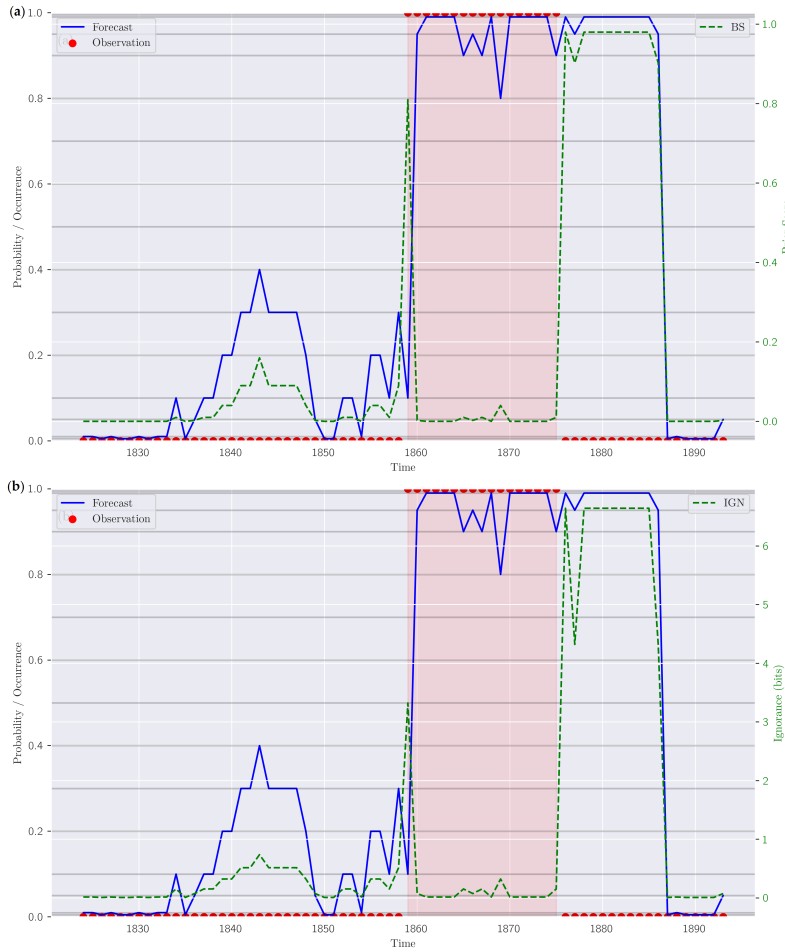

**Figure 2.** Time series of forecasts (blue solid line) and observations (red circles at 0 and 1). The red colour-fill indicates the period during which the rare event is observed, where the x-axis is non-dimensional timesteps. The verification score (green dotted line) per timestep evaluates the old model against observations: panel (**a**) uses the Brier Score (unitless); panel (**b**) uses Ignorance, in bits.

There is a small difference between maximum BS values for the two zones of high BS, with the "probabilistic false alarm" of high (99.5%) probabilities punished harder in (2) due to the much lower (10–30%) probabilities issued during (1). This is intuitive: there is less of a distance between the predicted and observed probabilities. The magnitude of difference is less instinctive, though, and we note the punishment of (1) using IGN (Figure 2b) is around half the bits of ignorance than in zone (2). This is a difference of interpretation when analysing the same dataset, begging the question of which score provides a better estimate of skill.

Figure 3 shows the same time window but for forecasts from the new model. The main difference when analysing skill is a shorter time period of (evidently unhelpful) high probabilities after the event has finished, in comparison to the old model (Figure 2). There is now a different time series of BS and IGN values, and we observed a similar couplet of zones around the event where the new model gave false positives or negatives in a probabilistic sense. As with the old model, the use of BS suggests both zones are as skilful (penalised around equally) as each other, whereas IGN indicates the false positive in zone (2) is punished over half as strongly as in (1). The implication is that using BS to evaluate this event might mislead the scientist when considering the impacts of inaccurate probabilistic forecasts: from panel a, we may conclude addressing overconfidence in an event is worth as much as miscalibration of underconfidence. In contrast, when we consider the area under the IGN curve as an available error we can remove, IGN indicates a different character.

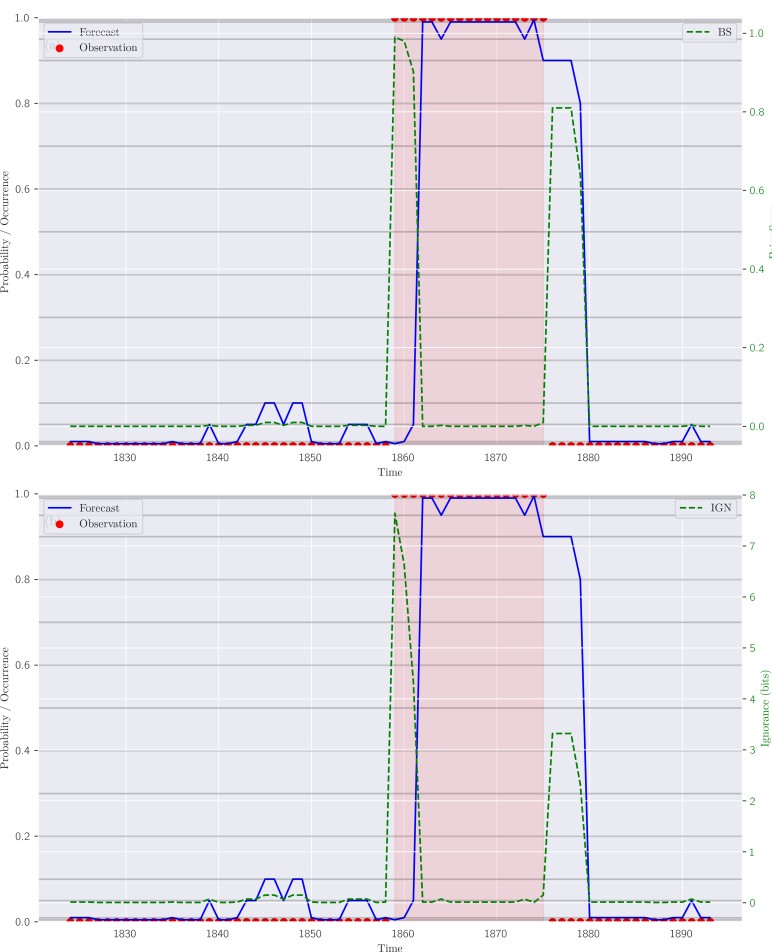

**Figure 3.** As Figure 2 but for the new model verification against the same observations.

In Figure 4, we demonstrate the issue with reliability diagrams with extremely rare events: insufficient data points in the centre for a better-performing new model. A dominance of zeros in both datasets reduces the usefulness of the reliability curve (blue line).

(The problem of zeros may be avoided by prescribing a minimum of one event observed per bin representing a Bayesian prior belief. In the figure, we leave the noisy dataset as-is as a demonstration.) We add bars of REL and DSC that represent the error generated from over-/under-confidence and knowledge gained from correct categorisation, respectively. We find the new model has less reliability error, and there is more uncertainty removed through discrimination skill (neglecting the noisy 0.5–0.7 bins, inclusive). To further explore the extremes, we look at the set of unique Ignorance values that result from permutations of probability bins *K* and the binary observation (Figure 5).

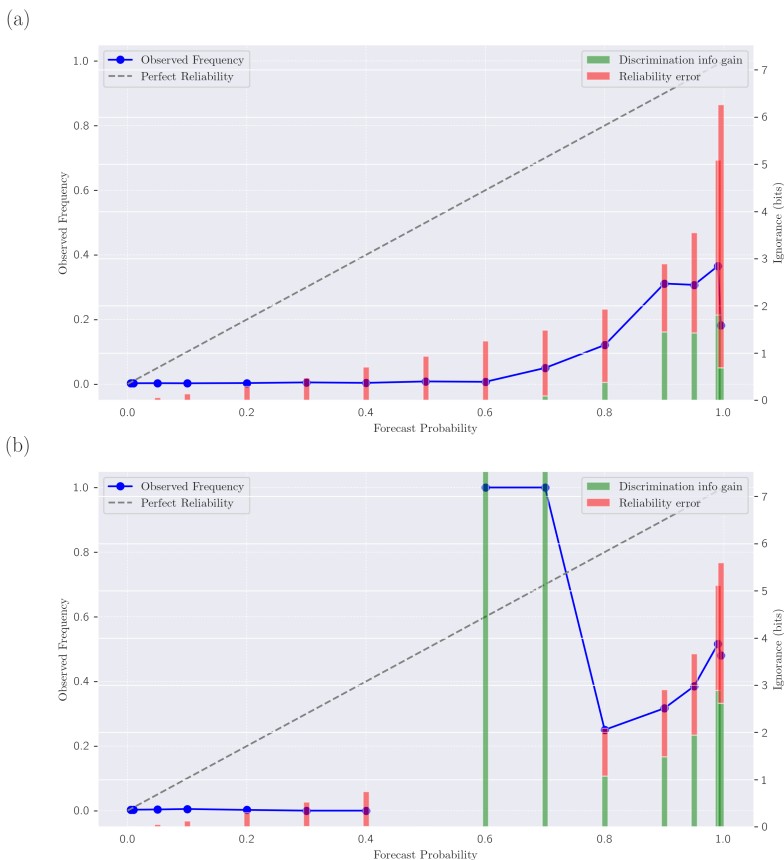

**Figure 4.** Reliability diagram, with forecast probability on the x-axis and frequency of occurrence on the y-axis. The panels show evaluation for the (**a**) old and (**b**) new models. The break in data points occurs in b due to insufficient samples to capture the full distribution.

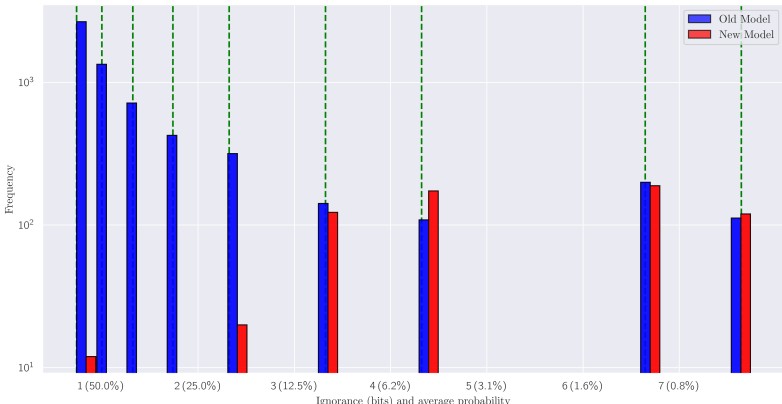

**Figure 5.** Histogram distribution of ignorance remaining (in bits) after each forecast for the old and new models, where only bins exceeding 0.7 bits are included to ignore the large proportion of near-zero-error forecast–observation pairs.

We only consider values greater than 0.7 bit to ignore the majority of the dataset that occurs with low probabilities and no event (representing common but barely useful forecasts). The values are all positive, measuring the remaining ignorance about the future event after a forecast is received. A larger value is a worse forecast, and the largest values occur when an event occurs with near-zero forecast probabilities, or the event does not occur when forecast with near-certainty. We find the old and new model alike suffer similar penalties over the dataset for their overconfidence. However, the new model accumulates much less error at the range of IGN values of 0.7–2.5. This fits with a comparison of Figures 2b and 3—the area of "probabilistic false alarm" is larger in the old model.

We can again use the decomposed scores of reliability (REL) and discrimination (DSC) to analyse the error character further. We form the skill scores IGNSS and BSS using Equation (15), where unity is a perfect forecast and zero is the no-skill line. The baseline used in the skill scores is the UNC component (i.e., the entropy of the observation base rate). Figure 6 depicts these scores for the old and new models. Information gain is represented in Figure 6 as the reduction in ignorance between the old and new models (or information loss, should the new model perform worse). For comparison, the reduction in BS represents the analogous concept. The sharp reduction in reliability error REL suggests the new model is more confident in making its correct forecasts; conversely, there is no gain in discrimination skill DSC, meaning all information gain is due to better calibration. Accordingly, the evaluator may now re-balance resources to address the low DSC skill. The skill scores IGNSS and BSS are unbounded to negative infinity, but we use the relative change to infer improvement from both scores. We find both scores indicate an improvement, but a larger improvement in BSS. Even when SS is negative, models may still have utility. The example in Figure 2 shows the double-penalty problem, where a single temporal error yields two doses of punishment. This undesired behaviour can be addressed by techniques such as reducing time series or gridded data to objects [53,54], or methods to allow for errors in time and space [17,55]. The rarer the event, the more data points required to estimate this event's frequency sufficiently, which means the double penalty occurs more often. We do not add a smoothing to the time series herein to avoid sensitivity to convolution or the windowing method—this is another source of subjectivity. However, while only the relative change is discussed herein, this does yield unfairly negative skill scores in Figure 6. This does not impact the interpretation of IGN and BS herein (although the magnitude of information transfer decreases due to smoothing, and hence less ignorance to begin with).

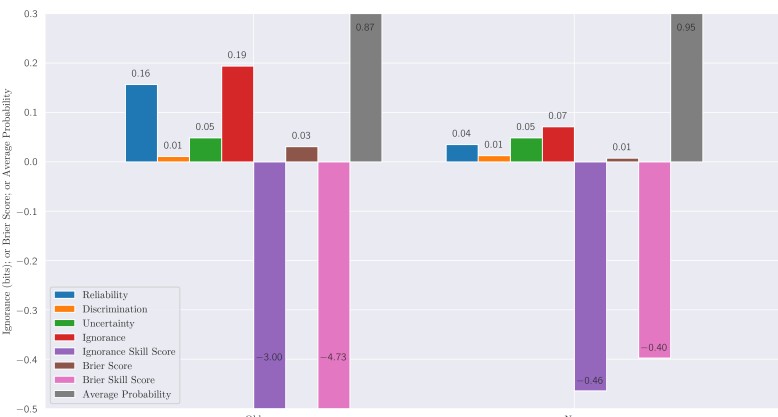

**Figure 6.** Components of ignorance for old and new models, with Brier-type scores for comparison. Information gain is represented as the reduction in ignorance (red) by using the new model over the old.

Ultimately, information is gained or lost by using the new model at each time-series sample instead of the old model. We show the distribution of information gain in Figure 7, magnifying the tails of the distribution as inset boxes. The distribution as a whole (main axis) shows a slight skew towards positive values (information gained over the old model).

The inset shows new model gains are likewise larger (the integrated area of the histogram bars; blue box) than the losses (red box) at larger information-content values.

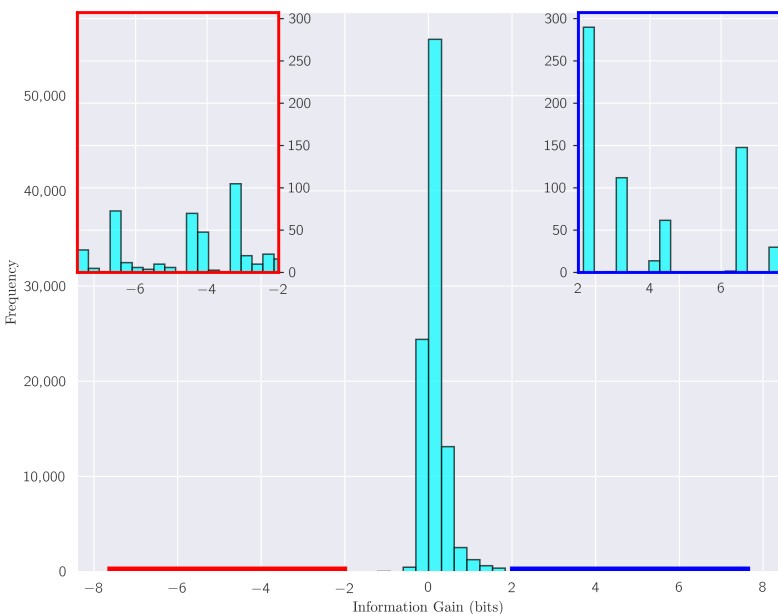

**Figure 7.** The histogram of information gained or lost by using the new model over the old. The upper-right inset axis, outlined in blue, has a smaller y-axis range, magnifying the histogram of information gain values greater than 2. This zoom is indicated by a blue line on the x-axis. Likewise, information loss greater than 2 bits (negative values less than −2) is shown on a larger scale by the upper-left red-bordered inset.

Information content, having strictly defined units of bits, lends itself to an analysis by distribution on a linear scale that has clear meaning regarding information gained. Likewise, change in Brier Score ΔBS is in the probability space, and can be visualised similarly (Figure 8) and intuitively. While the distribution shape is similar to Figure 7, it is difficult to interpret values explicitly as there is no inherent meaning in the difference of MSE of probabilities in the BS definition.

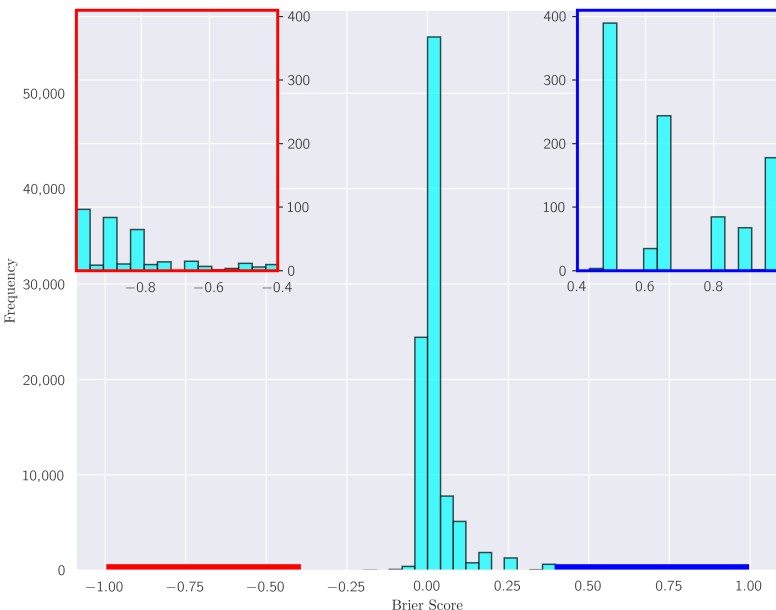

**Figure 8.** As Figure 7 but for the change in Brier Score ΔBS, with a magnification of the histogram at values greater than 0.4 and less than −0.4 of ΔBS.

## 6. Synthesis

Herein, we show how a scoring rule based on information theory is better suited to fairly evaluate forecasts of rare events, rather than scores derived from the Brier Score. This is of interest to meteorologists in a field with high-impact, low-frequency events that are difficult to capture by operational NWP models [51]. We conceptualise skill as the information gained by an improved forecast over a baseline, and compute information gain by the reduction in the Ignorance score. Further,

- Information gain accounts for higher moments of the probability distribution, as the Brier Score is equal to a second-order approximation of Ignorance. The Brier Score captures a mean error, but not more complex aspects of the probability distributions such as kurtosis.
- Differences between the two scores deviate sharply at extreme probabilities, such as those that characterise a dataset of rare events, since the BS truncates the score as probabilities tend to zero (rather than tending to infinity as in a logarithmic score);
- Information as measured in units of *bits* allows comparison between variables and joins a more universal framework of information transfer or surprise removal; the idea of optimising through information gain is shared with machine learning, for instance, and allows cross-comparison of system-component performance;
- We can decompose the gain or loss of information into reliability–discrimination components as with the Brier family of scores;
- While we laud objectively beneficial mathematical properties, information-based scores (unlike the Brier family) require the subjective clipping of certain forecast probabilities $f \in \{0,1\}$—binary events that do not verify yield an infinite error that reflects the implication of believing a prediction without question.

We showed above how visualising model performance can be interpreted similarly to Brier-type scores. The benefit of using the information framework is the preservation of more information at the extreme probabilities and a firmer meaning of the units of improvement between the models. Increased use of information gain would align the optimisation loss-function (scoring rule) for AI and meteorological models alike, especially as ML/AI is increasingly used in the atmospheric and earth sciences [56–60]. We encourage further work to implement information-theory metrics in a manner intuitive with which to forecast, straightforward to interpret during evaluation, and with a clear definition of what skill is measured. Information has a strict meaning. However, it is not a measure of utility: a high information content gained may not necessarily be especially practically useful.

The Brier Score occurs in other verification metrics: when BS is extended to multiple categories (i.e., $o \in \{0, 0.5, 1\}$), the BS can be extended to ranked (Ranked Probability Score; RPS) [61] and continuous ranked (CRPS) [62] scores. This is useful for evaluating a continuous or multicategory variable, such as precipitation. Unlike RPS and CRPS, the use of IGN instead of BS renders these ranked scores statistically *local*—and faster computationally [45]. The BS can be replaced with IGN for the assessment of multicategory forecasts [5]. We also find the BS within the Fractions Skill Score (FSS): this can be replaced with IGN as the scoring rule in FSS to create an "information fractions skill score", albeit with appropriately modified interpretations as discussed herein.

**Author Contributions:** Conceptualisation, J.R.L.; methodology, J.R.L.; software, J.R.L.; validation, J.R.L., C.K.P. and K.N.; formal analysis, J.R.L., C.K.P. and K.N.; investigation, J.R.L.; resources, J.R.L.; data curation, J.R.L.; writing—original draft preparation, J.R.L.; writing—review and editing, J.R.L., C.K.P. and K.N.; visualisation, J.R.L.; supervision, C.K.P.; project administration, J.R.L. and C.K.P.; funding acquisition, J.R.L. All authors have read and agreed to the published version of the manuscript.

**Funding:** J.R.L. is funded by Uintah County Special Service District 1 and the Utah Legislature.

**Data Availability Statement:** The latest code for the scores and figures used herein can be downloaded at: https://github.com/johnrobertlawson/info-gain-usu/blob/master/supplementary_jupy.

ipynb (accessed on 1 April 2024). The original contributions presented in the study are included in the article, further inquiries can be directed to the corresponding author.

**Acknowledgments:** The authors thank reviewers for improving the quality of the manuscript, and Harold Brooks for a preliminary review and discussion regarding the assessment of weather forecasts. J.R.L. thanks Jimmy Correia, Patrick Skinner, Leonard Smith, Seth Lyman, Dan White, Bart Wolf, and many others for inspiring conversation. Brainstorming with OpenAI GPT-4 output accelerated project development and helped link disparate concepts. GitHub Co-Pilot output was used to assist Python code development. No generative AI was used verbatim in the writing of this paper.

**Conflicts of Interest:** Author Kenric Nelson was employed by the company Photrek. The remaining authors declare that the research was conducted in the absence of any commercial or financial relationships that could be construed as a potential conflict of interest.

**Abbreviations**

The following abbreviations are used in this manuscript:

| | |
|---|---|
| IG | Information Gain |
| IGN(SS) | Ignorance (Skill Score) |
| B(S)S | Brier (Skill) Score |
| $f$ | Forecast probability scalar |
| $o$ | Observation (0 or 1) scalar |
| $K$ | Probability bins |
| REL | Reliability |
| DSC | Discrimination |
| UNC | Uncertainty |
| $D_{\mathrm{KL}}$ | Kullback–Liebler Divergence |

**Appendix A. Brier Score as an Approximation of Ignorance**

A more brief derivation is found in Benedetti [4]; here, we work through the demonstration in more steps for accessibility. In the case of a binary event,

$$IGN = D_{\mathrm{KL}}(o \parallel f) = o \log_2\left(\frac{o}{f}\right) + (1-o)\log_2\left(\frac{1-o}{1-f}\right) \tag{A1}$$

Now, we expand $D_{\mathrm{KL}}$ using a second-order Taylor series expansion around the point $o = f$. First, we find the first and second derivatives of $D_{\mathrm{KL}}$ with respect to $f$:

$$\frac{d}{df}D_{\mathrm{KL}}(o \parallel f) = \frac{-o}{f} + \frac{1-o}{1-f} \tag{A2}$$

$$\frac{d^2}{df^2}D_{\mathrm{KL}}(o \parallel f) = \frac{o}{f^2} + \frac{1-o}{(1-f)^2} \tag{A3}$$

Second, we expand around $o = f$ using the Taylor series up to the second order:

$$D_{\mathrm{KL}}(o \parallel f) \approx D_{\mathrm{KL}}(o = f) + \frac{d}{df}D_{\mathrm{KL}}(o = f) \cdot (o - f) + \frac{1}{2}\frac{d^2}{df^2}D_{\mathrm{KL}}(o = f) \cdot (o - f)^2 \tag{A4}$$

At $f = o$, both $D_{\mathrm{KL}}(o = f)$ and $\frac{d}{df}D_{\mathrm{KL}}(o = f)$ reduce to zero. Thus, the Taylor series simplifies to:

$$D_{\mathrm{KL}}(o \parallel f) \approx \frac{1}{2}\frac{d^2}{df^2}D_{\mathrm{KL}}(o = f) \cdot (o - f)^2 \tag{A5}$$

Substituting the value of $\frac{d^2}{df^2} D_{KL}(o = f)$ from above, we obtain (and simplify to):

$$D_{KL}(o \parallel f) \approx \frac{1}{2}\left(\frac{o}{o^2} + \frac{1-o}{(1-o)^2}\right) \cdot (o-f)^2 \tag{A6}$$

$$\approx \frac{1}{2}\left(\frac{1}{o} + \frac{1}{1-o}\right) \cdot (o-f)^2 \tag{A7}$$

As the Brier Score (BS) is defined in Eqn. 1, we can view $BS(o, f)$ as an approximation of $D_{KL}(o \parallel f)$:

$$D_{KL}(o = f) \approx \frac{1}{2}\left(\frac{1}{o} + \frac{1}{1-o}\right) \cdot BS(o = f) \tag{A8}$$

We find that $D_{KL}$ is equal to BS scaled by two arbitrary factors, as in Equation (3). Analogously for completeness, if we use the definition of IG in terms of $D_{KL}$, observations $o$, and two forecast catalogues $f_1$ and $f_2$,

$$IG(o, f_1, f_2) = \sum_{x \in \mathcal{X}} o(x) \log_2\left(\frac{o(x)}{f_1(x)}\right) - \sum_{x \in \mathcal{X}} o(x) \log_2\left(\frac{o(x)}{f_2(x)}\right) \tag{A9}$$

$$= a \cdot \left(BS(o, f_1) - BS(o, f_2)\right) + c \tag{A10}$$

here $a$ and $c$ are again arbitrary constants.

## Appendix B. Histogram Binning for Forecast Evaluation

From the time series, we allocate each pair of forecast and observed probabilities into bins and generate the reliability and discrimination ability of the forecasting system. If we have a set of probabilities from which $f$ is issued, rather than continuous values, we can skip the intermediate step of categorising the data into bins. If not, we generate a histogram, quantising the data as follows. Given a set of forecast probabilities, $\mathcal{F} = \{f_1, f_2, \ldots, f_n\}$, and corresponding observations, $\mathcal{O} = \{o_1, o_2, \ldots, o_n\}$, we define the bins into which these probabilities will be categorised:

$$\mathbf{K} = [k_1, k_2, \ldots, k_m] \tag{A11}$$

Each forecast probability $f_i$ is put into a bin $k$ to generate a fully categorised (quantised) time series. For each bin $k$, we count the number of forecasts ($n_k$) and the sum of observations ($o_k$) that fall into that bin.

$$n_k = \sum_{i=1}^{n} \mathbb{1}_{\{f_i = k\}} \tag{A12}$$

$$o_k = \sum_{i=1}^{n} o_i \cdot \mathbb{1}_{\{f_i = k\}} \tag{A13}$$

where $\mathbb{1}$ is the indicator function, returning 1 if the forecast $f_i$ falls into bin $k$ and 0 otherwise. After computing $n_k$ and $o_k$ for each bin $k$, we can now calculate inherent uncertainty, discrimination, and reliability, as outlined in Appendix C.

## Appendix C. Decomposition into Reliability, Discrimination, and Uncertainty

*Appendix C.1. Kullback–Leibler Divergence ($D_{KL}$)*

We start with a full expansion of $D_{KL}$. The $D_{KL}$ function, in the text as Equation (13), is fully expanded as:

$$D_{KL}(p \parallel q) = p \log_2\left(\frac{p}{q}\right) + (1-p) \log_2\left(\frac{1-p}{1-q}\right), \tag{A14}$$

for truth frequency $p$ and forecast probability $q$ in the application herein. The notation used above matches the typical symbols in information theory, where it represents the information lost when $q$ is used to approximate $p$.

*Appendix C.2. Uncertainty (UNC)*

The uncertainty metric UNC is the entropy of the overall observed event frequency ($\bar{o}$):

$$\text{UNC} = \begin{cases} 0 & \text{if } \bar{o} = 0 \text{ or } \bar{o} = 1, \\ -(\bar{o} \log_2 \bar{o} + (1 - \bar{o}) \log_2(1 - \bar{o})) & \text{otherwise.} \end{cases} \tag{A15}$$

This metric quantifies the inherent uncertainty in the observations on average, with higher values indicating greater uncertainty. If the event always or never occurs, there is no uncertainty; hence, the first choice in Equation (A15).

*Appendix C.3. Reliability (REL)*

The reliability metric REL is calculated as the expected $D_{\text{KL}}$ between observed frequencies and the forecast probabilities across all bins:

$$\text{REL} = \frac{1}{N} \sum_{k \in K} D_{\text{KL}}(\bar{o}_k, k) \cdot n_k, \tag{A16}$$

where $\bar{o}_k$ is the observed frequency in bin $k$, $k$ is the forecast probability bin, and $n_k$ is the count of forecasts in bin $k$. The $D_{\text{KL}}$ function measures the divergence between the observed and forecast probabilities, with larger values indicating less reliable forecasts.

*Appendix C.4. Discrimination (DSC)*

The discrimination metric DSC assesses the forecast's ability to differentiate between different outcomes. It is computed using the overall observed frequency ($\bar{o}$) as the reference:

$$\text{DSC} = \frac{1}{N} \sum_{k \in K} D_{\text{KL}}(\bar{o}_k, \bar{o}) \cdot n_k, \tag{A17}$$

where $D_{\text{KL}}(\bar{o}_k, \bar{o})$ quantifies the divergence between the observed frequency in each bin and the overall observed frequency. In our context, it measures the ignorance reduced by correct categorisation of the outcome, or inherent goodness.

*Appendix C.5. Full Expansion and Link with Brier Score and Cross-Entropy*

Hence, considering a time series of observation–forecast pairs $(\mathcal{O}, \mathcal{F})$, the full expansion of $D_{\text{KL}}$ (and likewise IGN) is

$$\begin{aligned} D_{\text{KL}}(\mathcal{O} \,\|\, \mathcal{F}) &= \text{UNC} - \text{DSC} + \text{REL} \tag{A18} \\ &= -(\bar{o} \log_2 \bar{o} + (1 - \bar{o}) \log_2(1 - \bar{o})) \\ &\quad - \frac{1}{N} \sum_{k \in K} D_{\text{KL}}(\bar{o}_k, \bar{o}) \cdot n_k \\ &\quad + \frac{1}{N} \sum_{k \in K} D_{\text{KL}}(\bar{o}_k, k) \cdot n_k, \tag{A19} \end{aligned}$$

For reference, this is the parallel Brier Score decomposition with identical notation [1,5,18]:

$$\text{BS}(\mathcal{O} \,||\, \mathcal{F}) = \text{BS}_{\text{UNC}} - \text{BS}_{\text{DSC}} + \text{BS}_{\text{REL}} \tag{A20}$$

$$= \bar{o}(1 - \bar{o})$$

$$- \frac{1}{N} \sum_{k \in K} (\bar{o}_k - \bar{o})^2 \cdot n_k$$

$$+ \frac{1}{N} \sum_{k \in K} (k - \bar{o}_k)^2 \cdot n_k, \tag{A21}$$

The Cross-Entropy (CE) loss or score [13] is a scoring measure related to $D_{\text{KL}}$ that permits forecasts evaluated against uncertain observations (i.e., $o \in [0, 1]$). The CE score can be decomposed similarly [11]. We prefer to use $D_{\text{KL}}$ to measure skill in the present paper as the observation-uncertainty characteristics are not required. The difference in using CE is the addition of uncertainty in the observations; however, with binary verification, there is no uncertainty, and CE simplifies to $D_{\text{KL}}$. This is a straight comparison with BS, which does not consider observation error (i.e., $o \in \{0, 1\}$). Note the difference in uncertainty of the observation dataset (UNC and $\text{UNC}_{\text{BS}}$) versus uncertainty in assigning events to what was actually observed (zero if we have full confidence in observations).

**Appendix D. Generating a Synthetic Time Series**

The observed series is generated based on a predefined probability of thunderstorm occurrence $\bar{o}$. We lay out the methodology below to better correspond to the Python code provided with this manuscript. For each time $i$ in the series of length $N$, the occurrence of the event $o$ is determined by:

$$o_i = \begin{cases} 1 & \text{if } U(0, 1) < \bar{o} \\ 0 & \text{otherwise} \end{cases} \tag{A22}$$

where $U(0, 1)$ represents a random draw from a uniform distribution over the interval $[0, 1]$. Autocorrelation is introduced to the observed series to simulate temporal dependency seen in real-world meteorological data. This is achieved by potentially copying the occurrence value from the previous day based on the autocorrelation factor ($R$), set as 0.8 herein:

$$o_i = \begin{cases} o_{i-1} & \text{if } U(0, 1) < R \\ o_i & \text{otherwise} \end{cases} \tag{A23}$$

The forecast probabilities are initialised and adjusted based on the observed series and a maximum forecast error ($E$). We set the error constant for the old model $E_{old}$ as 3.33 $E_{new}$ to create an inferior forecast set. The adjustment incorporates $\bar{o}$ and a normally distributed error term to mimic forecast uncertainty:

$$f_i = \begin{cases} \bar{o} + \epsilon & \text{if } o_{i-1} = 0 \\ 1 - \bar{o} + \epsilon & \text{if } o_{i-1} = 1 \end{cases} \tag{A24}$$

where $\epsilon \sim N(0, E^2)$ represents the forecast error. Autocorrelation is similarly applied to the forecast series, adjusting subsequent forecasts based on previous values:

$$f_i = \begin{cases} f_{i-1} + \epsilon & \text{if } U(0, 1) < R \\ f_i & \text{otherwise} \end{cases} \tag{A25}$$

Finally, to ensure that forecast probabilities remain within the valid range [0, 1], they are clipped accordingly:

$$f' = \begin{cases} min(K) & \text{if } f \leq 0, \\ max(K) & \text{if } f \geq 1, \\ f & \text{otherwise.} \end{cases} \tag{A26}$$

This process discretises continuous forecast probabilities into predefined bins *K*. Forecasts with probabilities of 0% and 100% are adjusted to the smallest and largest probability bins to enable correct binning (and avoid divergence to infinity). Each forecast probability $f'$ is then assigned to the nearest bin using the *digitize* function from the python *numpy* package. The forecast probabilities are then replaced with their corresponding bin values, quantising the continuous probabilities into discrete categories.

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
