# Peer review of "Decoding the Atmosphere: Optimising Probabilistic Forecasts with Information Gain"

_2674-0494, doi:10.3390/meteorology3020010_

Round 1
Reviewer 1 Report
Comments and Suggestions for Authors
This paper sets out the potential benefit of applying information-based scores to probabilistic forecasts, and contrasts with the more traditional Brier-type scores. The method is well set out and the examples used are appropriate to make the case argued. This paper will be of interest and use in assessing probabilistic forecasts.
What is the comparative advantage of using an information-based approach (compared to traditional approaches) in verifying forecasts?
This question is critical to improving forecasts and better understanding when any why there are more or less useful. This is important and interesting.
This is not the first application of information-based approaches to verification, but such approaches are not widely used or appreciated, so this is novel and original in that respect. This work adds an appreciation of the shortcoming of conventional approaches to verification and offers a useful alternative.
The paper well well-written and the text is clear and easy to read. The conclusions are consistent with the evidence and arguments presented. The authors address the main question posed.
Author Response
Thank you for your time reviewing the manuscript!
Reviewer 2 Report
Comments and Suggestions for Authors
This paper explores evaluation of probabilistic atmospheric prediction when the event of interest is rare with information theory. The paper is well written. Several aspcts need to be improved.
1) The space before the paragrahs should be removed in several places, e.g. Lines 62, 82, please find similar places.
2) The equations should be properly numeraterd. where is eq. (17)?
3) The conclusions shold be properly summarized in a separate section.
4) The proportion of recent literatures should be enhanced.
Author Response
Thank you for your time and effort reviewing this manuscript.
- Thanks for catching those errors with indents and ghost equation references.
- We've added a few more references.
- We appreciate the suggestions regarding a conclusions section. However, we do not consider our paper worthy of a distinct section: rather, we compromise in putting the bulleted list to highlight the main points.
Reviewer 3 Report
Comments and Suggestions for Authors
The article “Decoding the Atmosphere: Optimising Probabilistic Forecasts with Information Gain” discusses the optimization of probabilistic forecasts to get more relevant information, especially for rare events. These types of studies are add-ons to climate modeling where the forecast generated can be represented as information gain. The study has mostly used a set of Brier scores and information theory to achieve the goal of Information Gain in terms of reducing Ignorance.
The article has nicely portrayed the advantage of using information theory-based skill score to compare different probabilistic forecasts and the magnitude of improvement/ deterioration over each other. The article is well written and the methods are clearly presented for readers.
Minor Comment:
It would have been better if authors could have showed the results using real forecast from ECMWF/NOAA.
Author Response
Thank you for your time and effort in this review. We appreciate your comment about real-life data, for which our paper's scope was unfortunately too narrow. One author is indeed following this route with a project proposal evaluating NWS forecasters.